# Dietary Supplementation of Shredded, Steam-Exploded Pine Particles Decreases Pathogenic Microbes in the Cecum of Acute Heat-Stressed Broilers

**DOI:** 10.3390/ani11082252

**Published:** 2021-07-30

**Authors:** Akshat Goel, Beom-June Kim, Chris-Major Ncho, Chae-Mi Jeong, Vaishali Gupta, Ji-Young Jung, Si-Young Ha, Dong-Hwan Lee, Jae-Kyung Yang, Yang-Ho Choi

**Affiliations:** 1Department of Animal Science, Gyeongsang National University, Jinju 52828, Korea; genesakshat@gnu.ac.kr (A.G.); premix@naver.com (B.-J.K.); major159@gnu.ac.kr (C.-M.N.); wjdcoa@gnu.ac.kr (C.-M.J.); vaishali2020@gnu.ac.kr (V.G.); 2Institute of Agriculture and Life Sciences, Gyeongsang National University, Jinju 52828, Korea; charmhanjjy@nate.com (J.-Y.J.); jkyang@gnu.ac.kr (J.-K.Y.); 3Division of Applied Life Sciences (BK21 Plus Program), Gyeongsang National University, Jinju 52828, Korea; 4Department of Environmental Materials Science, Gyeongsang National University, Jinju 52828, Korea; hellohsy2@gmail.com (S.-Y.H.); ehdxos@naver.com (D.-H.L.)

**Keywords:** acute heat stress, broilers, cecum metagenome, insoluble fiber, performance, steam-exploded pine

## Abstract

**Simple Summary:**

The importance of the gut in poultry can be explained by the microbiome action. Prebiotics has gained attention as potential substances for improving gut health. The presence of insoluble dietary fiber from a cheap source such as wood specially prepared by the steam explosion that facilitates the depolymerization of hemicellulose could be an added advantage for microbial existence. Heat stress (HS) has been known to have drastic effects on chickens. In this study, we investigated the effect of dietary steam-exploded pine particle (SPP) supplementation and subsequent acute HS on productive performance and cecum microbiome in broilers. The HS tends to decrease the percent difference in body weight and rectal temperature. Metagenome analysis revealed similar richness and diversity in microbial communities. Firmicutes and Bacteroidetes were the most abundant phylum and were inversely correlated with each other. Furthermore, Firmicutes was also inversely correlated with unfavorable bacterial phyla. Supplementation of SPP in diets helped by enhancing favorable and reducing unfavorable bacterial genus in the cecum of the HS chickens. Although a clear advantage of using SPP on production parameters in broilers was not revealed, this study provided useful information to understand the modulation of microbiota during HS in dietary SPP supplemented chickens.

**Abstract:**

The gut microbiome stimulates nutrient metabolism and could effectively generate heat tolerance in chickens. This study investigates the effects of dietary steam-exploded pine particle (SPP) supplementation and subsequent acute heat stress on productive performance and cecum microbiome in broilers. Eight-day Ross 308 broilers were distributed in three groups with 0%, 1%, and 2% SPP in diets. On the 41st day, forty birds were allocated to four groups with ten birds each. The treatments were control diet at thermoneutral temperature (0% NT) and acute heat-stressed (HS) birds fed control (0% HS), 1% (1% HS), and 2% (2% HS) SPP. Parameters recorded were body weight (BW), feed intake (FI), rectal temperature (RT), relative organ weight, and metagenome analysis from cecum samples. Percent difference in BW, FI, and RT was decreased in HS birds. Metagenome analysis revealed similar richness and diversity in microbial communities. The relative abundance of the bacterial genus such as *Limosilactobacillus*, *Drancourtella*, and *Ihubacter* was increased while that of *Alistipes*, *Alkalibacter*, *Lachnotalea*, and *Turicibacter* was decreased in SPP supplemented HS birds. Concludingly, the production performance of broilers is negatively influenced during HS, and 2% dietary SPP supplementation may reduce the adverse effects of HS by modifying the microbiota in chickens.

## 1. Introduction

The consistent increase in the environmental temperature is a serious threat to livestock production. Broiler production contributes significantly to the livestock industry and is severely affected under high ambient temperature conditions. Heat stress (HS) leads to drastic effects on the production performances and immunocompetence that enhances the mortality rate in chickens [1,2]. Various dietary supplements have been utilized to attempt to minimize the drastic effects of HS. However, due to the extensive competitive nature of the poultry industry, it is critical to keep feed costs at the lower end. The utilization of agricultural waste in the context of the cheap availability of additives could be one of the strategies for selection. Furthermore, choosing a feed additive with heat mitigating properties may provide extra benefits.

Prebiotics are non-digestible fibers that improve health by positively modifying the favorable intestinal microflora [3]. Microorganisms present in the gut have a crucial role in maintaining intestinal homeostasis [4]. Supplements that contribute to modulating the microflora can be of great importance. In light of the above facts, using non-digestible fibers from low-cost agriculture byproducts could be a good solution. Wood powder is rich in non-digestible fibers and can act as the source for the growth of the good microorganisms in the gut, and can be used as a prebiotic.

Further processing of wood powder from different sources such as pine trees may enhance its benefits. Thermal treatments may increase the utilization of insoluble dietary fiber sources from wood. Among the different existing thermal treatments, steam explosion, for example, is one of the most commonly used for the fractionation of biomass components. Steam explosion pretreatment is a simple, low-cost, and environmentally friendly technology that causes the depolymerization of hemicellulose and lignin into soluble oligomers, taking advantage of its high-temperature profile [5]. Thus, steam explosion facilitates the breakdown of lignocellulosic biomass structure [6,7] and has been used to apply their end products to a variety of purposes such as feed [8,9,10].

The gut is a vital organ of the digestive system, and most of the undigested fibrous material reaches the large intestine. It has previously been demonstrated that the breakdown of fibrous material takes place in the cecum by anaerobic fermentation of undigested fibrous material such as cellulose, starch, and other resistant polysaccharides through microbial action [11]. Thus, the importance of evaluating the microbial load in the cecum is high. Dietary supplementation of additives with prebiotic properties may help the proliferation of useful microbes and simultaneously reduce the propagation of pathogens in the gut [12]. However, due to the scarce use of wood powder from a pine tree in chicken diets, little information is available regarding its modulatory role on the gut microbiota of chickens, especially under HS conditions.

The present study evaluated the effect of increasing amounts of dietary supplementation of pine tree (*Pinus densiflora*) particles on the performance and cecum microbiota in acute heat-stressed broiler chickens. We hypothesize that birds fed on diets containing pine tree particles may increase good bacteria, reduce the pathogenic load in the cecum during HS conditions, and further help mitigate the harmful effects of HS on production performances in chickens. 

## 2. Materials and Methods

The present experiment was conducted at the animal research facility of the Gyeongsang National University, Korea. The Animal Ethics Committee approved all the experimental procedures of the Gyeongsang National University (GNU-200916-C0057). 

### 2.1. Animal Housing and Treatments

In the study, 260-day-old straight run (mixed sex) Ross 308 broiler chicks were procured from a local hatchery (Ohsung Hatchery, Seongju, Korea) and raised in a controlled environment with continuous lighting. For the first seven days, all chicks were distributed in a total of thirteen cages, with 20 chicks in each cage. A commercial feed and water were supplied ad libitum. On the 8th day, each chick was weighed (*n* = 216) and assigned to one of three different treatment groups containing twelve replicates of each treatment and six birds in each cage. The experimental diets contained 0%, 1%, and 2% shredded, steam-exploded pine particles (SPP) passing through a 10-mesh sieve replacing corn in their feed ingredients. The selection of dosage was made based on the results of the previous studies [10]. The preparation of SPP was done by exploding the pinewood chips of approximately 2 × 2 × 0.5 cm^3^ with steam at 200 °C for 11.5 min and stored at 20 °C until use. On the 41st day of age, a total of 40 birds (0%: 20 birds, 1%: 10 birds, 2%: 10 birds) were randomly selected and distributed into four groups of five replicates each treatment and two birds per cage. One group (0% SPP) was kept at thermoneutral temperature (21.0 °C) served as control while the other three groups (0, 1, and 2% SPP) were heat-stressed in a separate room by gradually increasing the temperature of the room to 31 °C within the first three hours and then maintained the same temperature for another 3 h. The total HS period was 6 h (Figure 1). Finally, there were a total of four treatments: Control diet (0% SPP) at thermoneutral temperature (0% NT); Control diet with acute HS (0% HS), 1% SPP-supplemented diet at acute HS (1% HS), and 2% SPP-supplemented diet at acute HS (2% HS).

### 2.2. Sampling and Data Collection

At the end of the experiment, all the birds were weighed using digital balance, and rectal temperature (RT) was recorded by inserting a digital thermometer (HI 91610, Hanna instruments Inc., Padova, Italy) to approximately 3 cm inside the cloaca. Residual feed from each cage was also recorded to calculate the net feed intake. A total of 6 birds from each treatment were randomly selected and euthanized using carbon dioxide on 41 days of age. The liver, bursa of Fabricius, and spleen were dissected free, weighed, and presented as absolute and relative to body weight (BW). Cecum samples were immediately collected in sterilized 50 mL falcon tubes and stored at −80 °C. 

### 2.3. Microbial Analysis

Total DNA containing the microbial communities from the cecum sample was extracted using a DNeasyPowerSoil Kit (Qiagen, Hilden, Germany) according to the manufacturer’s instructions, followed by quantification using Quant-IT PicoGreen (Invitrogen, Waltham, MA, USA). To evaluate the metagenome of the cecum DNA samples, a 16S metagenomic sequencing library was constructed using a Herculase II Fusion DNA Polymerase Nextera XT Index Kit V2 (Illumina, San Diego, CA, USA) following manufacturer instruction. The library was then sequenced with the Illumina platform at Macrogen, Inc. (Seoul, Korea). FASTQ files were created for each sample which was then subjected to quality profiling, adapter trimming, and read filtering using fastp program [13]. The paired-end reads were assembled into one sequence using FLASH (v1.2.11) software [14], and the assembled sequences less than 400 bp and more than 500 bp were removed. The number of operational taxonomic units (OTUs) was determined by de novo clustering with a 97% sequence identity cutoff using the CD-HIT-EST program [15]. Each sequence OTU was checked for its taxonomic similarity using BLAST+ (v2.9.0) program [16] against the reference database (NCBI 16S Microbial), and the identical coverage of less than 85% was identified as not defined. The microbial communities in each sample in terms of OTU abundance and taxonomic information were analyzed using QIIME (v1.9) software. The species diversity and homogeneity among the microbial community in a sample were evaluated through Shannon, Goods Coverage, and Inverse Simpson Index. Alpha diversity was presented through the rarefaction curve of the Chao1 and observed OTUs. Beta diversity was evaluated using Weighted/Unweighted UniFrac distances.

### 2.4. Statistical Analysis

The data relating to animal studies such as growth performances, RT, absolute and relative organ weights were analyzed using one-way ANOVA, followed by Duncan’s multiple range test. Alpha diversity (community richness and diversity) and taxonomic analysis (phylum and genus) were done using the Kruskal–Wallis test and adjusted with Bonferroni correction. Differences were considered statistically significant at *p* < 0.05 unless otherwise stated. All the data were expressed as mean ± standard error of the mean (SEM). The beta diversity analysis includes principal coordinate analysis (PCoA), calculated by weighted Unifrac and unweighted Unifrac. Spearman’s correlation was used to evaluate the correlation among the bacteria of the phylum. Analysis was conducted using IBM SPSS Statistics package 25.0 (IBM software, Chicago, IL, USA). Graph pad Prism software was used to draw the graphs.

## 3. Results

There was no difference in the initial and final BW of chickens measured before and after six hours of HS. The percent difference in BW was significantly affected, and HS birds had a lower (*p* < 0.05) percent difference in BW irrespective of diets compared to birds kept at control temperature. Feed intake was also decreased (*p* < 0.05) in dietary SPP (1% and 2% SPP) supplemented HS birds in comparison to birds kept at thermoneutral temperature (Table 1).

Compared with those in the birds kept at thermoneutral temperature, rectal temperature (RT) was significantly increased (*p* < 0.001) in HS birds having 0, 1, or 2% SPP in diets, indicating an HS effect (Table 2).

The absolute and relative organ weights of the liver, bursa of Fabricius, and spleen were similar, and no differences (*p* > 0.05) were observed among different treatments (Table 3). 

A total of 24 cecum samples (six each treatment group) were used to generate 574,031 sequences ranging from 19,503 to 29,051 sequences for each sample after a 97% sequence similarity and removal of chimeric reads for quality control. A total of 5522 operational taxonomic units (OTU) were generated via clustering analysis with an abundance greater than 0.005% ranged from 196 to 260 OTUs for each sample. 

Rarefaction curves for chao1 and observed OTUs (Figure 2) became flattered to the right and reached a plateau indicating that a reasonable number of reads were used in the analysis and reveals most of the bacterial community in the cecum samples of chickens. 

Community richness was analyzed using OTUs and Chao1 and is presented in Figure 3. The OTUs and Chao1 were similar and were not affected among the treatment groups. The diversity in the community was analyzed using Shannon, Inverse Simpson, and Goods coverage and is presented in Figure 3. Shannon index has shown increasing trend (*p* = 0.071) in 2% HS and 0% HS against its counterparts. No variation was observed in Inverse Simpson and Goods coverage and was similar among the treatments group. 

The PCoA based on unweighted unifrac distances were similar, while weighted unifrac distances showed scattered plots (Figure 4). 

The phylum microbiota of cecum samples from different treatments is presented in Figure 5. At the phylum level the cecum microbiota was dominated by Firmicutes (0% NT: 60.48%, 0% HS: 65.35, 1% HS: 70.40, 2% HS: 73.17) and Bacteroidetes (0% NT: 34.58%, 0% HS: 31.99, 1% HS: 27.62, 2% HS: 24.86) followed by Verrucomicrobia, Tenericutes, Actinobacteria, Proteobacteria and Candidatus Melainabacteria. 

The two predominant phyla, Firmicutes and Bacteroidetes are strongly correlated inversely (Spearman R = −0.950, *p* = 0.001) in terms of their abundances (Figure 6). Similar inverse correlation of Firmicutes was also observed with Verrucomicrobia (Spearman R = −0.525, *p* = 0.008) and Proteobacteria (Spearman R = −0.366, *p* < 0.079). However, Firmicutes was positively correlated (Spearman R = 0.397, *p* = 0.054) and Bacteroidetes was inversely correlated (Spearman R = −0.427, *p* = 0.038) with Actinobacteria.

The Firmicutes to Bacteroidetes ratio has shown an increasing trend with increasing concentration of SPP (Figure 7).

Figure 8 presents the cecum microbiota at the genus level in different treatment groups. The analysis of genus microbiota revealed that *Bacteroides*, *Faecalibacterium*, and *Blautia* are the most dominant genera followed by *Limosilactobacillus*, *Alistipes*, *Lactobacillus*.

Among the top five dominant genera, *Alistipes* and *Limosilactobacillus* were significantly different among treatments. The relative abundance of *Alistipes* was decreased (*p* = 0.010), and that of *Limosilactobacillus* (*p* = 0.042) was increased by 2% HS in comparison to 0% HS (Figure 9). Five more genera were significantly modified by increasing the dietary concentration of SPP and subsequent HS in the cecum of broiler chickens (Figure 9). For instance, the abundance of *Ihubacter* was increased (*p* = 0.049) in 2% HS compared to 1% HS while the abundance of *Alkalibacter* was decreased (*p* = 0.003) in 1% HS and 2% HS in comparison to 0% NT. The abundance of *Lachnotalea* was increased (*p* = 0.049) in 0% HS compared to 0% NT but was similar and showed numerically lower values in 2% HS compared to 0% NT. The abundance of *Drancourtella* was increased (*p* = 0.039) in 2% HS in comparison to 0% NT. The abundance of *Turicibacter* was increased (*p* = 0.042) in 0% HS in comparison to 2% HS. However, not significant, but a numerically higher abundance of *Turicibacter* was also seen in 0% HS than that of 0% NT.

## 4. Discussion

Broilers are meat-type chickens and are mostly reared for livestock production. Therefore, continuous improvement in the performance of broilers is the prerequisite of the poultry industry. Several stressors were identified that negatively influence broiler performances. HS is among the environmental stressors known to have drastic effects on production performance in chickens. The present study was conducted to evaluate the adverse effect of acute HS and to identify if supplementation of SPP during HS can lead to overcoming some adverse effects of HS. Although no effect was observed on the initial and final BW of the broilers among the treatment groups, a decrease in the percent difference in BW in the HS birds suggested the harmful effects of HS on chickens. A decrease in the percent difference in BW could be attributed to a reduction in feed intake. An increase in respiration rates [17] due to the absence of sweat glands may reduce the feed intake of birds. Furthermore, secretion of anorexic hormone under HS could also result in decreasing feed intake [18]. The present study also reported a decrease in feed intake in HS (1 and 2% SPP) birds. This corroborates the previous studies where acute HS resulted in decreasing the BW gain and feed intake when birds were exposed to 31 °C for ten h [19]. Contrary to this, in the present study, no variation was found in the feed intake of birds kept at thermoneutral or HS when fed a 0% SPP diet. The difference in the results could be attributed to the time of HS for which it was conducted. 

High ambient temperature tends to increase the RT in chickens. In the present study, RT was increased in birds exposed to HS in comparison to birds kept at thermoneutral temperatures. Our results are consistent with the previous reports where the HS challenge at 35 °C even for 1.5 and 3 h enhances RT in chicken [20]. An increase in the RT could be related to respiration rates. Due to the absence of sweat glands, chickens are less able to dissipate heat in the surroundings, thus increasing respiration rates [17], which may further increase RT.

Different organs have specific roles in the body. The liver is vital for nutrition and metabolism in the body, whereas the bursa of the Fabricius and spleen are mainly responsible for imparting immunity in chickens. Stress is known to induce modulation in organ development, but inflection in weight will depend on the severity of stress to which the chickens are exposed. The weights of the liver, bursa of Fabricius, and spleen were similar and not affected among the treatment groups. These results correlate with the previous study where lymphoid organ weights (thymus, spleen, and bursa of Fabricius) were not affected when birds were exposed to 31 °C for 10 h [19]. Contrary to this, a decrease in the lymphoid organ weight was reported when birds were exposed to 31 °C or 36 °C for ten h from 35 days to 41 days of age [1]. Together, the difference in the results could be related to the time of exposure and the intensity of HS.

The progression of microbes in the gut starts as early as after hatch and may be influenced by many factors such as diets and other environments. It has been suggested that dietary prebiotic supplementation may influence growth performance by positively modifying the gut microbiota in chickens [21]. Since prebiotics cannot be absorbed or digested in the upper gastrointestinal tract (GIT), it passes to the lower GIT where it acts as a source of food for the good bacteria reducing the pathogenic bacteria attachment. However, HS has been associated with the degradation of intestinal integrity and GIT impairment, leading to the penetration of pathogenic microbes in chickens [22]. It is expected that dietary SPP supplementation as a source of prebiotic may help in reducing the adverse effect of HS due to the growth of beneficial bacteria and reduction of pathogenic bacteria.

To analyze the community richness, OTUs and Chao1 estimators were evaluated, and no difference was observed among the treatment groups. This suggests that the number of species in all the treatment groups was similar and was not affected by either HS or SPP supplementation. A similar trend was also observed in estimating the community diversity as Inverse Simpson, and Goods coverage was not affected among the treatment groups. However, the Shannon index did show an increasing trend in 2% HS and 0% HS treatments. The SPP acts as a prebiotic and may help in increasing the abundance of beneficial bacteria in the gut [12], thus showing increasing trends of microbial diversity in 2% HS treatment. The increasing trend of Shannon diversity in 0% HS treatment could be attributed to the invasion of pathogenic microbes due to challenged intestinal gut integrity under HS [22]. 

In general, unweighted and weighted unifrac distances are used in microbial ecology, accounting for the presence or absence of observed organisms and their abundance, respectively. The PCoA based on unweighted unifrac distances showed scattered plots indicating similar microbiota. However, for weighted unifrac distances, dimension one contributed 48.04% of the inertia while dimension two contributed 18.69%, cumulatively contributing to almost 66% variability in caecum microbiota of SPP supplemented thermoneutral and heat-stressed chickens. Higher variability in weighted unifrac distances could be attributed to the addition of SPP-containing fiber in diets which may enhance the abundance of microflora in the cecum in a concentration-dependent manner. In the present study, Firmicutes and Bacteroidetes were found to be the most abundant microbiota in the chicken cecum, which was similar to the previous studies [23]. Furthermore, our results also showed that these two most abundant phylum microbiotas are inversely correlated with each other, and the Firmicutes percentage increases while Bacteroidetes percentage decreases with increasing concentration of dietary SPP supplementation. This could be related to the negative correlation of Bacteroidetes with BW along with higher cecum Firmicutes to Bacteroidetes ratio in obese hosts [24]. Although we did not find any significant differences in BW and Firmicutes to Bacteroidetes ratio, increasing Firmicutes to Bacteroidetes ratio trends with increasing concentration of dietary SPP supplementation indicates fattening of chicken due to the prebiotic effects. The reason behind the nonsignificant results could be related to the short term of HS. Long-term HS has been associated with growth retardation in chickens [1]. Higher differences with significantly positive variations in the SPP treatment groups could be expected if HS persists for an extended period of time and opens the way for future studies. The Bacteroidetes phylum was also found to be inversely correlated with another phylum of Actinobacteria. The exact reason behind this not clear. However, it could be due to the positive trend among the correlation between Firmicutes and Actinobacteria. A similar trend was also reported in chickens on the 9th and 18th day of age after hatching [25].

The bacteria of phylum Verrucomicrobia are known to possess mucin degrading ability and, if found in abundance, are associated with several diseases [26]. Proteobacteria is another gram-negative bacteria from the phylum that has been associated with pathogenicity and inflammation [27]. Verrucomicrobia was found to be negatively correlated with the Firmicutes in the present study, and its number was decreased with the increasing concentration of SPP supplementation in HS birds. A similar negative correlation trend was also observed among Firmicutes and Proteobacteria. However, a numerically higher number of Firmicutes was found with increasing SPP supplemented chickens. This indicates that SPP supplementation may help in maintaining gut health by reducing the colonization of bacteria associated with the degradation of intestinal barrier integrity and thus reducing the penetration of pathogenic microbes.

Due to a lack of information about the specific role of *Alistipes* in chickens, Biasato and his group suggested importance for its characterization, but they looked upon its increased abundance in their treatment group as a positive effect [28]. Parker and his group discussed the protective effects of *Alistipes* against liver fibrosis, colitis, cancer immunotherapy, and cardiovascular disease and suggested its pathogenic effects in colorectal cancer and mental depression [29]. However, in the present study, the relative abundance of *Alistipes* was decreased by 2% HS in comparison to 0% HS. However, the exact reason behind this is not clear since *Alistipes* belongs to phylum Bacteroidetes which has shown a decreasing trend with increasing dietary SPP concentration in the present study and could be the reason for the significant reduction of *Alistipes*.

*Limosilactobacillus reuteri* produces antimicrobials, including organic acids, ethanol, and reuterin, obstructing the pathogenic microbial colonization [30]. Numerically lower values of *Limosilactobacillus* in 0% HS in comparison to 0% NT might be due to the enhanced penetration of pathogens under HS. Furthermore, a significant increase in the *Limosilactobacillus* abundance in 2% HS in comparison to 0% HS proposes the beneficial effects of SPP supplementation under HS conditions. The presence of higher *Limosilactobacillus* may toughen the intestinal barrier and reduces the chances of pathogen penetration. 

Gut microbiota has been associated with the production of trimethylamine *N*-oxide (TMAO), having a protective effect against adverse conditions such as temperature, salinity, and hydrostatic pressure [31]. Although its increased levels have been associated with the adverse effect on cardiovascular health in humans [32], it has an important physiological role in lower animals [31]. In chickens, the preferred level of TMAO is not yet evaluated and its adverse effects on cardiovascular health are yet to be determined. *Ihubacter massiliensis* is one of the critical bacteria responsible for the production of TMAO [33]. In the present study, the abundance of *Ihubacter* was numerically higher in 2% HS against its counterparts (0% NT and 0% HS) and was significantly increased in 2% HS compared to 1% HS. We assume that SPP supplementation may have a protective effect on heat-stressed chickens by modifying the abundance of TMAO producing *Ihubacter*. We did not evaluate the TMAO levels in chicken, however, its correlation with *Ihubacter* to modify heat tolerance capacity could be interesting for future studies.

The abundance of *Alkalibacter* was significantly decreased in dietary SPP supplemented chickens (1% HS and 2% HS) in comparison to 0% NT. Prebiotic selectively stimulates the growth of beneficial bacteria, enhancing the production of short-chain fatty acids and lactic acid as fermentation products, reducing the intestine’s pH [34,35]. *Alkalibacter* being alkaliphilic might be reduced in the cecum of SPP supplemented chickens due to its prebiotic effect. Although the effect of HS in decreasing the abundance of *Alkalibacter* in this study could not be ignored, the relationship between the increase in temperature with pH is yet to be established. 

High ambient temperature produces oxidative stress and is associated with ROS generation [22]. Malondialdehyde (MDA) is the secondary product generated after the degradation of lipids during lipid peroxidation [36]. Thus, the MDA is expected to be increased under HS. However, the increase in MDA differs in tissues and depends on many factors such as intensity, time, and temperature of HS exposure. The abundance of *Lachnotalea glycerini* is positively correlated to serum MDA in mice [37]. An increase in *Lachnotalea* in 0% HS in comparison to 0% NT indicates the effect of HS. Furthermore, decreasing trend with numerically lower abundance of *Lachnotalea* in the 2% HS in comparison to 0% HS suggested the role of SPP supplementation to recover from HS. 

*Drancourtella* has been isolated from fresh human stool [38] and its role is not clear; however, an increase in its abundance in 2% HS in comparison to 0% NT indicates its beneficial effects in reducing the harmful effect of HS in the chicken cecum. Due to lack of information, future studies are warranted to confirm its role in the chicken gut.

*Turicibacter* is considered harmful for health due to its inverse correlation with tight junction in mice [39]. The presence of *Turicibacter* in monogastric animals may cause subclinical infection and can negatively modulate the microbiota of the gut [40]. The contagious activity of *Turicibacter* may perhaps be due to the intestinal damage caused by the pathogenic invasion. The increment in the abundance of *Turicibacter* in *Salmonella* infected chickens also supports the above notion [4]. The association of challenged gut health under HS conditions has already been explained previously [22]. The present study shows a numerically higher abundance of *Turicibacter* in 0% HS in comparison to 0% NT indicates the occurrence of pathogenic bacterial invasion due to the intestinal injury under HS. Significant reduction in the *Turicibacter* abundance in 2% HS treatment confirms the role of SPP supplementation as prebiotics to retain intestinal health and maintaining microbial homeostasis.

## 5. Conclusions

In conclusion, acute HS negatively influences broiler chickens’ production performance and rectal temperature. HS tends to increase the Shannon diversity index indicating pathogenic microbial filtration. Nevertheless, increased Shannon diversity index in dietary SPP supplemented chickens along with fewer pathogenic phylum indicates improved gut health. An increase in the abundance of the favorable genus such as *Limosilactobacillus* and *Ihubacter* while decreasing unfavorable genus such as *Lachnotalea and Turicibacter* in SPP supplemented diets during HS suggested its role in modifying gut health.

## Figures and Tables

**Figure 1 animals-11-02252-f001:**
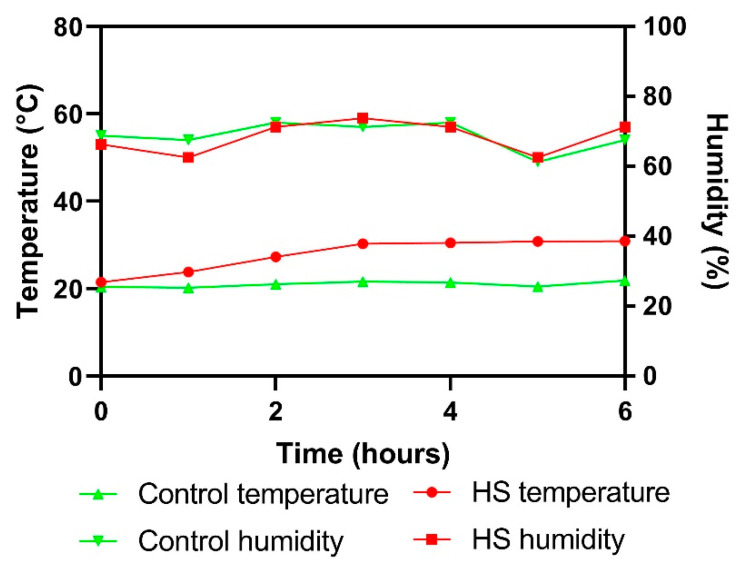
The temperature of thermoneutral control and heat stress rooms during the period of the experiment.

**Figure 2 animals-11-02252-f002:**
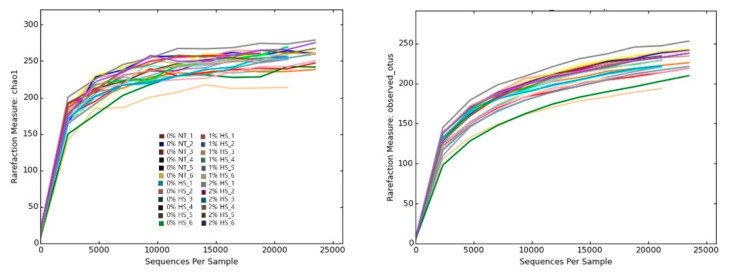
Rarefaction curve of Chao1 and observed OTUs for each sample. There are twenty-four samples represented by different colored lines that belong to four treatment groups of six replicates each. The treatments were control diet containing 0% steam-exploded pine particles (SPP) at thermoneutral temperature (0% NT), control diet with acute heat stress (0% HS), 1% SPP-supplemented diet at acute heat stress (1% HS), and 2% SPP-supplemented diet at acute heat stress (2% HS). The thermoneutral birds were maintained at 21.0 °C while the temperature of the heat-stressed room was raised to 31 °C within the first three hours and then maintained for another three hours that made the total HS period of six hours.

**Figure 3 animals-11-02252-f003:**
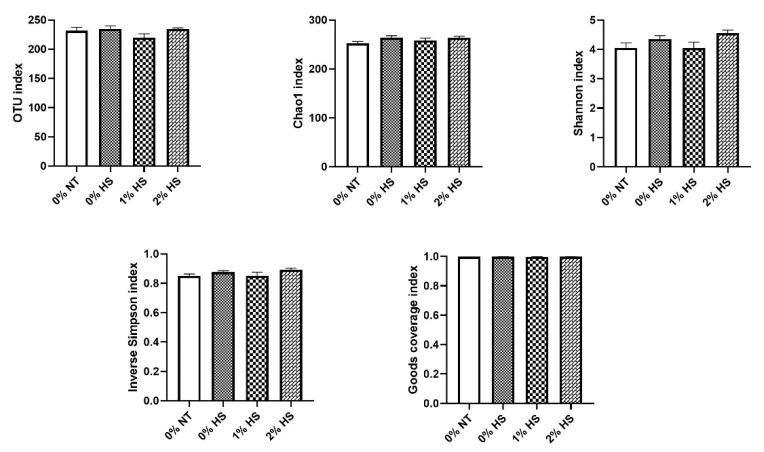
Effects of dietary shredded, steam-exploded pine particles (SPP) supplementation on the community richness and diversity of cecum microflora in broiler chickens exposed to either thermoneutral or heat stress conditions. The treatments were control diet containing 0% SPP at thermoneutral temperature (0% NT), control diet with acute heat stress (0% HS), 1% SPP-supplemented diet at acute heat stress (1% HS), and 2% SPP-supplemented diet at acute heat stress (2% HS). The thermoneutral birds were maintained at 21.0 °C while the temperature of the heat-stressed room was raised to 31 °C within the first three hours and then maintained for another three hours that made the total HS period of six hours.

**Figure 4 animals-11-02252-f004:**
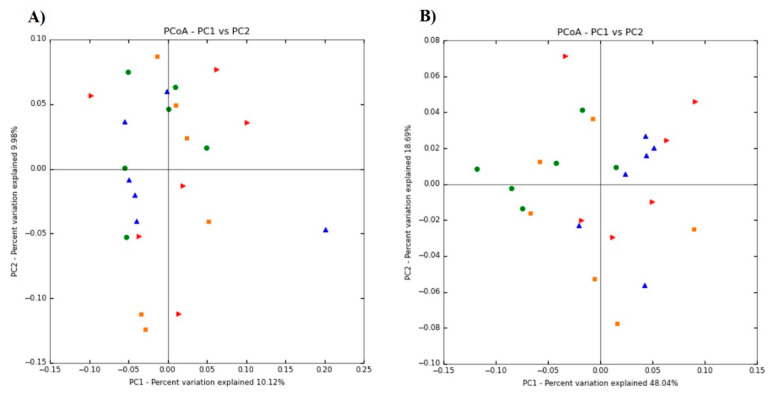
The composition changes of cecum microbiota based on principal coordinate analysis (PCoA) including unweighted (**A**) and weighted (**B**) unifrac distances in the dietary steam-exploded pine particles (SPP) supplemented chickens kept at either thermoneutral or at heat-stress. The treatments were control diet containing 0% SPP at thermoneutral temperature (0% NT), control diet with acute heat stress (0% HS), 1% SPP-supplemented diet at acute heat stress (1% HS), and 2% SPP-supplemented diet at acute heat stress (2% HS). The thermoneutral birds were maintained at 21.0 °C while the temperature of the heat-stressed room was raised to 31 °C within the first three hours and then maintained for another three hours that makes the total HS period of six hours. The red triangle indicates 0% NT, the blue triangle indicates 0% HS, the orange square indicates 1% HS, and the green circle indicates 2% HS treated chickens.

**Figure 5 animals-11-02252-f005:**
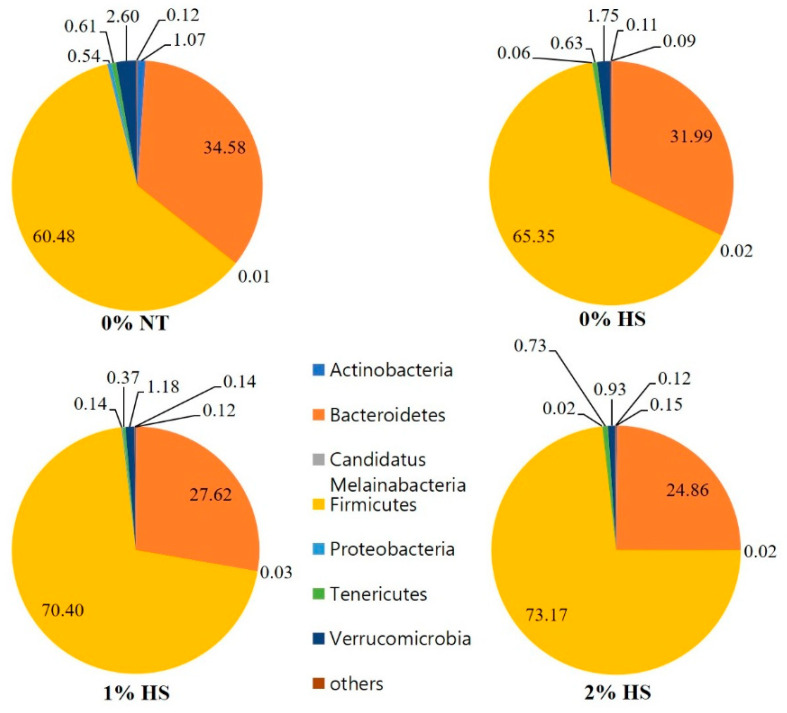
Effects of dietary steam-exploded pine particles supplementation to the chickens kept at either thermoneutral or heat stress on the composition of cecum microflora at Phylum level. The treatments were control diet containing 0% steam-exploded pine particles (SPP) at thermoneutral temperature (0% NT), control diet with acute heat stress (0% HS), 1% SPP-supplemented diet at acute heat stress (1% HS), and 2% SPP-supplemented diet at acute heat stress (2% HS). The thermoneutral birds were maintained at 21.0 °C while the temperature of the heat-stressed room was raised to 31 °C within the first three hours and then maintained for another three hours that makes the total HS period of six hours.

**Figure 6 animals-11-02252-f006:**
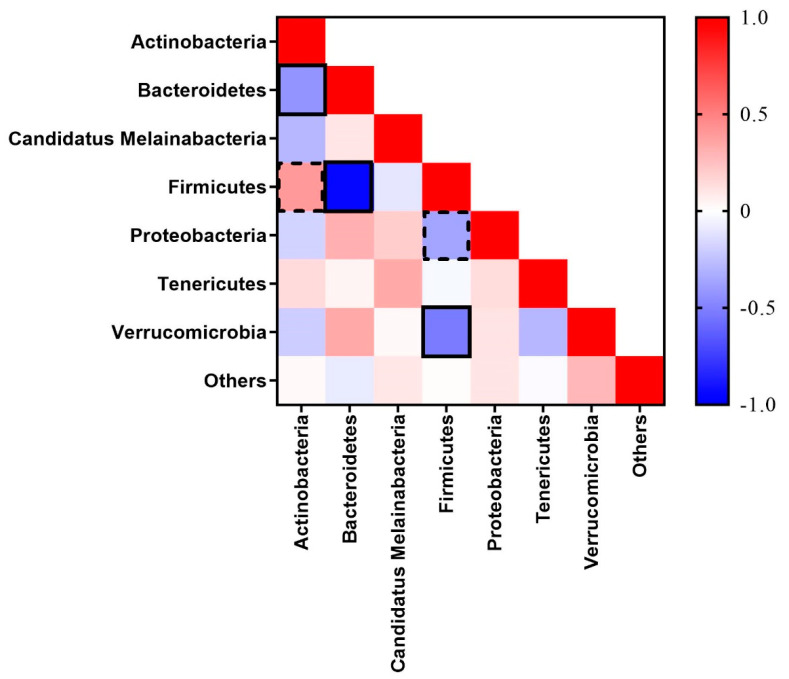
Correlation among the bacterial phylum of the cecum in broiler chickens supplemented with steam-exploded pine particles and kept at either thermoneutral or heat stress conditions. Values from each phylum were used to determine the correlation among each other. Zero indicates similarity without any possibility of correlation, while plus and minus values indicate positive and negative correlation, respectively. The continuous line and dash line indicate significance at *p* < 0.05 and *p* < 0.10, respectively.

**Figure 7 animals-11-02252-f007:**
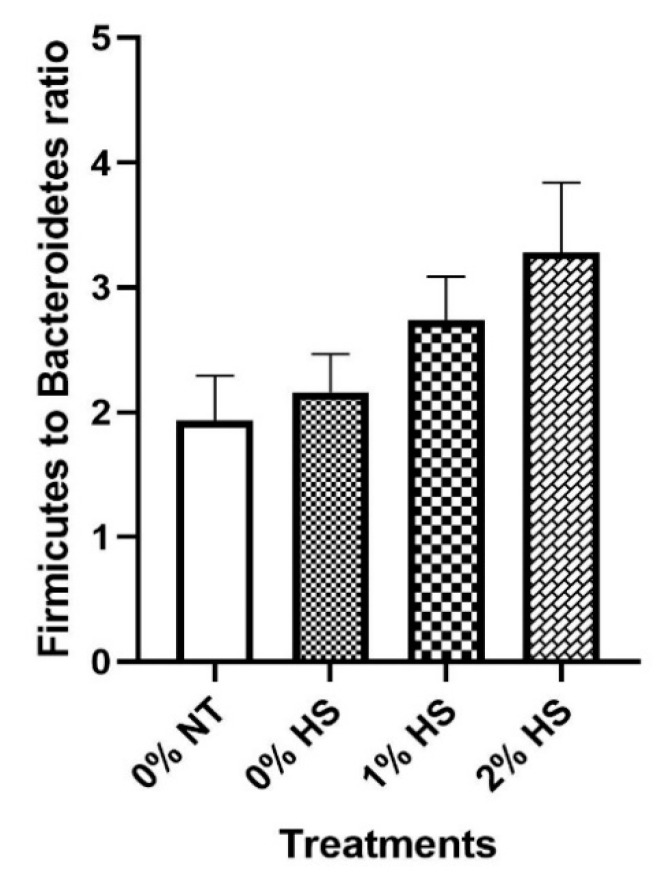
Effects of dietary steam-exploded pine particles supplementation on the Firmicutes to Bacteroidetes ratio in broiler chickens kept at either thermoneutral or heat stress conditions. The treatments were control diet containing 0% steam-exploded pine particles (SPP) at thermoneutral temperature (0% NT), control diet with acute heat stress (0% HS), 1% SPP-supplemented diet at acute heat stress (1% HS), and 2% SPP-supplemented diet at acute heat stress (2% HS). The thermoneutral birds were maintained at 21.0 °C while the temperature of the heat-stressed room was raised to 31 °C within the first three hours and then maintained for another three hours that makes the total HS period of six hours.

**Figure 8 animals-11-02252-f008:**
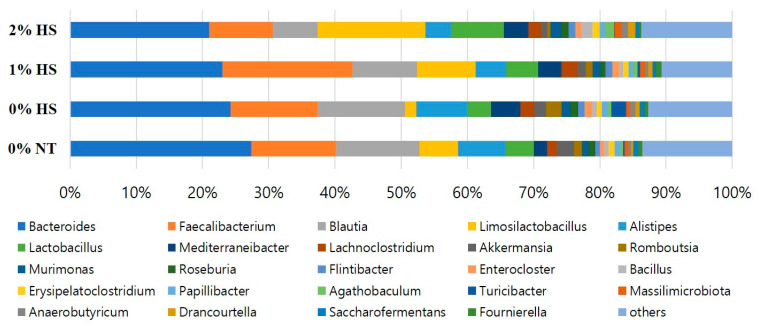
Effects of dietary steam-exploded pine particles supplementation and subsequent heat stress on the composition of cecum microflora of chicken at the genus level. Data represent the top 24 abundant microflorae. The treatments were control diet containing 0% steam-exploded pine particles (SPP) at thermoneutral temperature (0% NT), control diet with acute heat stress (0% HS), 1% SPP-supplemented diet at acute heat stress (1% HS), and 2% SPP-supplemented diet at acute heat stress (2% HS). The thermoneutral birds were maintained at 21.0 °C while the temperature of the heat-stressed room was raised to 31 °C within the first three hours and then maintained for another three hours that makes the total HS period of six hours.

**Figure 9 animals-11-02252-f009:**
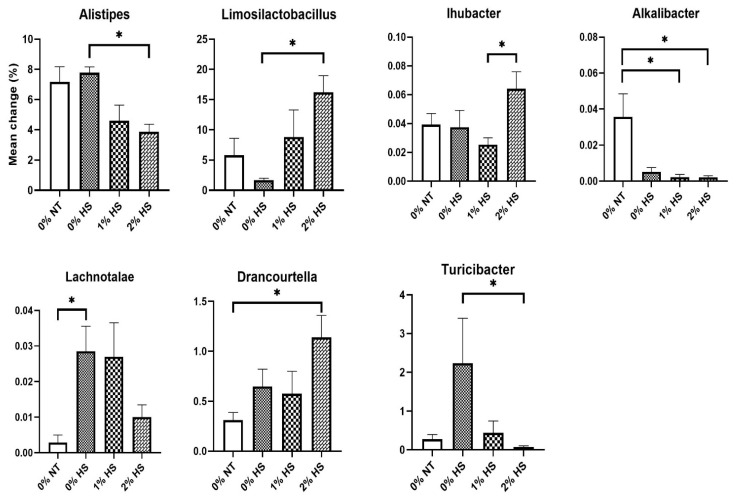
Effects of dietary steam-exploded pine particles supplementation and subsequent heat stress on the significantly modified genus of the cecum in broilers. The treatments were control diet containing 0% steam-exploded pine particles (SPP) at thermoneutral temperature (0% NT), control diet with acute heat stress (0% HS), 1% SPP-supplemented diet at acute heat stress (1% HS), and 2% SPP-supplemented diet at acute heat stress (2% HS). The thermoneutral birds were maintained at 21.0 °C while the temperature of the heat-stressed room was raised to 31 °C within the first three hours and then maintained for another three hours that makes the total HS period of six hours. * indicates a significant difference (*p* < 0.05).

**Table 1 animals-11-02252-t001:** Effects of dietary steam-exploded pine particles supplementation on the growth performances of thermoneutral and heat-stressed broiler chickens.

Treatments	Initial BW	Final BW	% Difference in BW	Feed Intake
0% NT	2689.5 ± 86.5	2721.5 ± 84.2	1.23 ^b^ ± 0.37	55.60 ^a^ ± 2.2
0% HS	2631 ± 111.2	2608.5 ± 109.8	−0.87 ^a^ ± 0.45	43.20 ^a,b^ ± 5.2
1% HS	2645.5 ± 39.7	2631.5 ± 39.5	−0.52 ^a^ ± 0.73	40.00 ^a^ ± 5.9
2% HS	2670 ± 50.7	2648.5 ± 49.7	−0.80 ^a^ ± 0.24	38.60 ^a^ ± 2.1
*p*-value	0.952	0.746	0.022	0.044

Chickens were fed with diets containing 0% (control), 1%, and 2% shredded, steam-exploded pine particles (SPP) from 8th day to 41st day of age. On 41st day, birds were either kept at thermoneutral temperature (21.0 °C) and provided control diet (0% NT) or heat-stressed at 31.0 °C for six hours and supplemented with 0% (0% HS), 1% (1% HS) and 2% (2% HS) SPP in diets. Data show mean ± SEM (*n* = 10). ^a,b^: different letters indicate significant differences (*p* < 0.05). Abbreviations: BW, body weight; NT, normal temperature; HS, heat stress.

**Table 2 animals-11-02252-t002:** Effects of dietary steam-exploded pine particles supplementation on the rectal temperature of thermoneutral and heat-stressed broiler chickens.

Treatments	Rectal Temperature	
	Before	After
0% NT	41.73 ± 0.03	41.88 ^a^ ± 0.05
0% HS	41.87 ± 0.04	43.75 ^b^ ± 0.25
1% HS	41.78 ± 0.05	43.52 ^b^ ± 0.23
2% HS	41.80 ± 0.04	43.30 ^b^ ± 0.20
*p*-value	0.157	0.001

Chickens were fed with diets containing 0% (control), 1%, and 2% shredded, steam-exploded pine particles (SPP) from the 8th day to the 41st day of age. On 41st day, birds were either kept at thermoneutral temperature (21.0 °C) and provided control diet (0% NT) or heat-stressed at 31.0 °C for six hours and supplemented with 0% (0% HS), 1% (1% HS) and 2% (2% HS) SPP in diets. “Before” and “After” indicates rectal temperature taken before starting and after completing six hours of heat stress respectively. Data show mean ± SEM (*n* = 6). ^a,b^: Different letters indicate significant differences (*p* < 0.05). Abbreviations: NT, normal temperature; HS, heat stress.

**Table 3 animals-11-02252-t003:** Effects of dietary steam-exploded pine particles supplementation on the absolute organ weight (g) and relative organ weight (% body weight) of thermoneutral and heat-stressed broiler chickens.

Treatments	Absolute Organ Weight (g)			Relative Organ Weight (%)		
	Liver	Bursa of Fabricius	Spleen	Liver	Bursa of Fabricius	Spleen
0% NT	65.0 ± 5.56	5.02 ± 0.44	2.92 ± 0.25	2.42 ± 0.18	0.19 ± 0.02	0.11 ± 0.01
0% HS	62.9 ± 2.92	4.18 ± 0.34	3.38 ± 0.61	2.46 ± 0.16	0.16 ± 0.01	0.13 ± 0.02
1% HS	62.3 ± 4.33	4.18 ± 0.63	3.44 ± 0.29	2.43 ± 0.16	0.16 ± 0.03	0.14 ± 0.01
2% HS	69.2 ± 3.60	4.57 ± 0.56	3.00 ± 0.46	2.59 ± 0.08	0.17 ± 0.02	0.11 ± 0.02
*p*-value	0.653	0.614	0.805	0.844	0.811	0.651

Chickens were fed with diets containing 0% (control), 1%, and 2% shredded, steam-exploded pine particles (SPP) from 8th day to 41st day of age. On 41st day, birds were either kept at thermoneutral temperature (21.0 °C) and provided control diet (0% NT) or heat-stressed at 31.0 °C for six hours and supplemented with 0% (0% HS), 1% (1% HS) and 2% (2% HS) SPP in diets. Data show mean ± SEM (*n* = 6). Abbreviations: NT, normal temperature; HS, heat stress.

## Data Availability

The data presented in this study are available on request from the corresponding author. The data are not publicly available due to other manuscripts under review.

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
