# Peer review of "Dietary Supplementation of Shredded, Steam-Exploded Pine Particles Decreases Pathogenic Microbes in the Cecum of Acute Heat-Stressed Broilers"

_animals, 2021, doi:10.3390/ani11082252_

Round 1

Reviewer 1 Report

My suggestion to improve the discussion is use Dunnett test to compare the SPP inclusion with the control group and Duncan to compare the experimental treatments. Reduce the discussion about the bacteria profile in cecum.

All my comments are in the text.

Author Response

Reviewer 1

Reviewers suggestion: My suggestion to improve the discussion is use Dunnett test to compare the SPP inclusion with the control group and Duncan to compare the experimental treatments. Reduce the discussion about the bacteria profile in cecum.

All my comments are in the text.

Authors reply: Authors are thankful to the reviewer for providing valuable suggestions to our manuscript.  The authors have addressed all the points raised by the reviewer, as shown below.  As per the reviewer’s suggestion we also tried the Dunnett test, but the results were similar, and since we did not find any differences, so we decided to keep the present analysis in the manuscript.  Discussion related to bacteria is also reduced as per the reviewer’s suggestion. 

Reviewers suggestion: Line 56 Minimize

Authors reply: The line 56 has been corrected as shown below:

Line 56: ‘Various dietary supplements have been tried to minimize the drastic effects of HS.’

Reviewers suggestion: Line 72 transformation of hemicellulose and lignin or only lignin into soluble oligomers?

Authors reply: It has been corrected according to the reviewer’s suggestion.

Lines 72-74: ‘Steam explosion pretreatment is a simple, low-cost, and environmentally friendly technology that causes the depolymerization of hemicellulose and lignin into soluble oligomers, taking advantage of its high-temperature profile [5].’

Reviewers suggestion: Line 154 maybe authors should try other test, like dunnett for example to compare with the control treatment. I think the results of growth performance would make sense.

Authors reply: As per the reviewer’s suggestion, we ran the Dunnett test and found similar results.  So, we have decided to go with the present statistical analysis in this manuscript.  The P values of the Dunnett test are presented below:

Treatments

Initial BW

Final BW

% difference in BW

Feed intake

0% NT Vs 0% HS

P > 0.05 (NS)

P > 0.05 (NS)

P < 0.05

P > 0.05 (NS)

0% NT Vs 1% HS

P > 0.05 (NS)

P > 0.05 (NS)

P < 0.05

P < 0.05

0% NT Vs 2% HS

P > 0.05 (NS)

P > 0.05 (NS)

P < 0.05

P < 0.05

Reviewers suggestion: Line 165 Delete this first phrase. 

Authors reply: The phrase has been deleted as per the reviewer’s suggestion.

Lines 168-170:Table 1 shows the effects of acute heat stress on growth performance after supplementing an increasing concentration of dietary SPP.  There was no difference in the initial and final BW of chickens measured before and after six hours of HS.’

Reviewers suggestion: Line 171 temperature (Table 1).

Authors reply: Line 171 has been corrected as per the reviewer’s suggestion.

Lines 173-174: ‘Feed intake was also decreased (p < 0.05) in dietary SPP (1% and 2% SPP) supplemented HS birds in comparison to birds kept at thermoneutral temperature (Table 1).’

Reviewers suggestion: Line 171 Table 1 try dunnett test to compare the experimental treatments with the control.

Authors reply: As per the reviewer’s suggestion, we tried the Dunnett test but  found similar results.  So, we have decided to go with the present statistical analysis in this manuscript.

Reviewers suggestion: Line 180 Delete the first phrase.

Line 181 Rectal temperature (RT).

Line 182 0% is not supplemented. The effect here is from HS only. Would be effect of HS despite the SPP supplementation.

Authors reply: The phrase has been deleted as per the reviewer’s suggestion. 

Lines 183-187:Table 2 presents the changes in RT before and after HS in different treatment groups.  RT was similar before HS and has no variation among treatment groups.  However, RT Compared with those in the birds kept at thermoneutral temperature, rectal temperature (RT) was significantly increased (p < 0.001) in HS birds having supplemented with 0, 1, or 2% SPP in diets, indicating a HS effect (Table 2).’

Reviewers suggestion: Line 193 3 and

Line 193 exclude the first phrase.

Line 193 weight, respectively,

Authors reply: As per the reviewer’s suggestion, the phrase has been deleted.  The corrections to the consecutive sentence have been presented below.

Lines 197-200:Tables 3, and 4 present the absolute and relative weight of the liver, bursa, and spleen.  The absolute and relative organ weights of the liver, bursa of Fabricius, and spleen were similar, and no differences (p > 0.05) were observed among different treatments  (Table 3).’

Reviewers suggestion: Line 194 I suggest only one Table with relative weight or if authors want to keep the absolute weight, put them in only one Table (absolute and relative weights).

Authors reply: Corrected as per reviewer’s suggestion, the two tables have been combined into one, as shown below:

Lines 201-208: ‘Table 3. Effects of dietary steam-exploded pine particles supplementation on the absolute organ weight (g) and relative organ weight (% body weight) of thermoneutral and heat-stressed broiler chickens. 

Absolute organ weight (g)

Relative organ weight (%)

Treatments

Liver

Bursa of

Fabricius

Spleen

Liver

Bursa of

Fabricius

Spleen

0% NT

65.0 ± 5.56

5.02 ± 0.44

2.92 ± 0.25

2.42 ± 0.18

0.19 ± 0.02

0.11 ± 0.01

0% HS

62.9 ± 2.92

4.18 ± 0.34

3.38 ± 0.61

2.46 ± 0.16

0.16 ± 0.01

0.13 ± 0.02

1% HS

62.3 ± 4.33

4.18 ± 0.63

3.44 ± 0.29

2.43 ± 0.16

0.16 ± 0.03

0.14 ± 0.01

2% HS

69.2 ± 3.60

4.57 ± 0.56

3.00 ± 0.46

2.59 ± 0.08

0.17 ± 0.02

0.11 ± 0.02

p-value

0.653

0.614

0.805

0.844

0.811

0.651

Chickens were fed with diets containing 0% (control), 1%, and 2% shredded, steam-exploded pine particles (SPP) from 8th day to 41st day of age.  On 41st day, birds were either kept at thermoneutral temperature (21.0 °C) and provided control diet (0% NT) or heat-stressed at 31.0 °C for six hours and supplemented with 0% (0% HS), 1% (1% HS) and 2% (2% HS) SPP in diets.  Data show mean ± SEM (n = 6).  Abbreviations: NT: normal temperature; HS: heat stress.’

Reviewers suggestion: Line 195 chicken (Table 3).

Authors reply: The line has been corrected as per the reviewer’s suggestion.

Lines 197-200: ‘The absolute and relative organ weights of the liver, bursa of Fabricius and spleen were similar, and no differences (p > 0.05) were observed among different treatments (Table 3).’

Reviewers suggestion: Line 203 This paragraph wouldn't be in the material and methods???

if the position is right, it should come after the table about viscera weights.

Authors reply: After deleting Table 4, this paragraph automatically comes after Table 3 having organ weights.  So, now the paragraph is according to the reviewer’s requirement and does not require any further action.

Reviewers suggestion: Line 475 Too much information about the bacteria. I suggest that authors sumarize this part with the bacteria that increased and reduced and the importance of them and of their concentration to the gut health.

Authors reply:  As per the reviewer’s suggestion, the information about the bacteria is shortened, making them more succinct, as shown below up to Lines 501-502. 

Lines 436-438: Alistipes are known to be a relatively new genus with the ability to resist bile and produce acetic acid due to carbohydrate fermentation [28].  Although scanty information is available on its role, different researchers have contradictory opinions.

Lines 449-452: ‘Limosilactobacillus is a new genus that has been reclassified from the genus Lactobacil-lus recently [31].  Limosilactobacillus reuteri is among the most abundant strain from this ge-nus that has been known to produces antimicrobials, including organic acids, ethanol, and reuterin, obstructing the pathogenic microbial colonization.’

Lines 458-463:Ihubacter is a recently classified genus containing Ihubacter massiliensis as an obligatory anaerobic, gram-negative, non-spore-forming bacteria [33]Gut microbiota has been associated with the production of trimethylamine N-oxide (TMAO), a small colorless amine oxide generated from choline metabolism, and has been suggested to have having a protective effect against adverse conditions such as temperature, salinity, and hydrostatic pressure [31].’

Lines 474-476:Alkalibacter is a gram-positive, alkaliphilic, rod-shaped, strictly anaerobic, and nonmotile bacterial genus from the family of Carnobacteriaceae with one known species Al-kalibacter saccharofermentans [37].

Lines 484-485:Lachnotalea is nonmotile, non-spore-forming slightly curved rods that are strictly an-aerobic in nature [40].

Lines 495-500:Drancourtella is an anaerobic, gram-positive, spore-forming, and nonmotile bacteria that has been isolated from fresh human stool [38] and its.  The exact role of Drancourtella is not clear; however, an increase in its abundance in 2% HS in comparison to 0% NT indicates its beneficial effects in reducing the harmful effect of HS in the chicken cecum.  Due to lack of information, future studies are warranted to confirm its role in the chicken gut.’

Lines 501-502:The genus Turicibacter are anaerobic, gram-positive, rod-shaped bacteria that are considered harmful for health due to their inverse correlation with tight junction in mice.’

Reviewers suggestion: Line 504 results in conclusion. Please, be concise in this item.

Authors reply:  As per the reviewer’s suggestion the concise results are presented and other information has been deleted from the text.

Lines 516-525: From this study, it can be In conclusion,ded that acute heat stress negatively influences the broiler chickens’ production performance and rectal temperature of broiler chickens.  Firmicutes and Bacteroidetes were the most abundant phylum.  Firmicutes were inversely correlated with pathogenic phyla such as Verrucomicrobia and Proteobacteria.  Heat stress tends to increase the Shannon diversity index indicating pathogenic microbial filtration.  Nevertheless, increased Shannon diversity index in dietary SPP supplemented chickens along with fewer pathogenic phylum indicates improved gut health.  An increase in the abundance of the favorable genus such as Limosilactobacillus and Ihubacter while decreasing unfavorable genus such as Lachnotalea and Turicibacter in SPP supplemented diets during HS suggested its role in modifying gut health.’

Reviewer 2 Report

1.Please provide the age of chicken at slaughter
2.Please provide the product information of steam-exploded pine particles
3.Why choose 1% and 2% steam-exploded pine particles
4.What daes "brusa" mean?plaease use " bursa of Fabricius"
5.Why did they only look at " liver, bursa, and spleen"?and none of them showed significant differences.
6.Suggest changing alpha diversity to a figure。

Author Response

Reviewers suggestion: 1. Please provide the age of chicken at slaughter

Authors reply: As per the reviewer’s suggestion, chicken age at the time of euthanizing has been included in the text.

Lines 125-127: ‘A total of 6 birds from each treatment were randomly selected and euthanized using carbon dioxide on 41 days of age.’

Reviewers suggestion: 2. Please provide the product information of steam-exploded pine particles

Authors reply: The information has been included as per the reviewer’s suggestion.

Lines 109-111: ‘The preparation of SPP was done by exploding the pinewood chips of approximately 2 x 2 x 0.5 cm3 with steam at 200 °C for 11.5 minutes and stored at 20 °C until use.’

Reviewers suggestion: 3. Why choose 1% and 2% steam-exploded pine particles

Authors reply:  The selection of dosage was made based on the feeding trial of previous studies conducted on a different source of wood in our laboratory.  As per the reviewer’s suggestion, the information has been included in the text and can be seen below.

Lines 108-109: ‘The selection of dosage was made based on the results of the previous studies [10].’

Reviewers suggestion: 4.What daes "brusa" mean?plaease use " bursa of Fabricius"

Authors reply: As per the reviewer’s suggestion, ‘bursa’ has been replaced with ‘bursa of Fabricius’ everywhere in the text.

Lines 127-128: ‘The liver, bursa of Fabricius, and spleen were dissected free, weighed, and presented as absolute and relative to body weight (BW).’

Lines 198-200: ‘The absolute and relative organ weights of liver, bursa of Fabricius and spleen were similar, and no differences (p > 0.05) were observed among different treatments (Table 3).’

Lines 201-208: ‘Table 3. Effects of dietary steam-exploded pine particles supplementation on the absolute organ weight (g) and relative organ weight (% body weight) of thermoneutral and heat-stressed broiler chickens. 

Absolute organ weight (g)

Relative organ weight (%)

Treatments

Liver

Bursa of

Fabricius

Spleen

Liver

Bursa of

Fabricius

Spleen

0% NT

65.0 ± 5.56

5.02 ± 0.44

2.92 ± 0.25

2.42 ± 0.18

0.19 ± 0.02

0.11 ± 0.01

0% HS

62.9 ± 2.92

4.18 ± 0.34

3.38 ± 0.61

2.46 ± 0.16

0.16 ± 0.01

0.13 ± 0.02

1% HS

62.3 ± 4.33

4.18 ± 0.63

3.44 ± 0.29

2.43 ± 0.16

0.16 ± 0.03

0.14 ± 0.01

2% HS

69.2 ± 3.60

4.57 ± 0.56

3.00 ± 0.46

2.59 ± 0.08

0.17 ± 0.02

0.11 ± 0.02

p-value

0.653

0.614

0.805

0.844

0.811

0.651

Chickens were fed with diets containing 0% (control), 1%, and 2% shredded, steam-exploded pine particles (SPP) from 8th day to 41st day of age.  On 41st day, birds were either kept at thermoneutral temperature (21.0 °C) and provided control diet (0% NT) or heat-stressed at 31.0 °C for six hours and supplemented with 0% (0% HS), 1% (1% HS) and 2% (2% HS) SPP in diets.  Data show mean ± SEM (n = 6).  Abbreviations: NT: normal temperature; HS: heat stress.’

Lines 366-368: ‘In general, the liver is vital for nutrition and metabolism in the body, while the bursa of Fabricius and spleen are mainly responsible for imparting immunity in chickens.’

Lines 370-371: ‘The weights of the liver, bursa of Fabricius, and spleen were similar and not affected among the treatment groups.’

Reviewers suggestion: 5.Why did they only look at " liver, bursa, and spleen"?and none of them showed significant differences.

Authors reply: The liver is a vital organ for nutrient metabolism.  Heat stress reduces feed intake and affects nutrient metabolism.  So, we wanted to see if liver weight is also modified and if SPP supplementation can modulate liver weight.  The bursa of Fabricius and spleen are responsible for immunity in chickens.  Heat stress is responsible for pathogenic penetration.  So, we wanted to evaluate if the weight is modified or not.  However, due to shorter duration and lower intensity of heat stress, these parameters were not affected.  We have explained this in the discussion section of the manuscript.

Lines 366-376: ‘Different organs have their specific roles in the body.  The liver is vital for nutrition and metabolism in the body, whereas the bursa of Fabricius and spleen are mainly responsible for imparting immunity in chickens.  Stress is known to induce modulation in organ development, but inflection in weight will depend on the severity of stress to which the chickens are exposed.  The weights of the liver, bursa of Fabricius, and spleen were similar and not affected among the treatment groups.  These results correlate with the previous study where lymphoid organ weights (thymus, spleen, and bursa of Fabricius) were not affected when birds were exposed to 31 °C for 10 h [19].  Contrary to this, a decrease in the lymphoid organ weight was reported when birds were exposed to 31 °C or 36 °C for 10 h from 35 days to 41 days of age [1].  Together, the difference in the results could be related to the time of exposure and the intensity of HS.’

Reviewers suggestion: 6. Suggest changing alpha diversity to a figure。

Authors reply: The table has been converted to figure as per the reviewer’s suggestion.  The figure is presented below.

Lines 240-249:

Figure 3. Effects of dietary shredded, steam-exploded pine particles (SPP) supplementation on the community richness and diversity of cecum microflora in broiler chickens exposed to either thermoneutral or heat-stress condition.  The treatments were control diet containing 0% SPP at thermoneutral temperature (0% NT), control diet with acute heat stress (0% HS), 1% SPP-supplemented diet at acute heat stress (1% HS), and 2% SPP-supplemented diet at acute heat stress (2% HS).  The thermoneutral birds were maintained at 21.0 °C while the temperature of the heat-stressed room was raised to 31 °C within the first three hours and then maintained for another three hours that made the total HS period of six hours.

Reviewer 3 Report

This is a very interesting paper.  Heat stress is a critical issue in terms of both production and animal welfare.  Any research that offers better management of heat stress is certainly needed.  Your pine particle work offers some valuable insight.  The Shannon index results are interesting.  Prebiotics are very popular today and this offers anther possible alternative with promising results.

Line 18 - replace "has" with "have"

Line 21 - add "s" to "chicken"

Line 52 - delete "The" to start the sentence

Line 56 - replace "encounter" with "offset"

Line 57 - "the" between "keep" and "feed"

Line 60 - replace "will" with "may"

Line 100 - were these chicks males, females, or straight runs?

Line 229 - add "s" to "groups"

Line 330 - remove "s" from "performances"

Line 331 - delete "the" between "on" and "production" 

Line 331 - remove "s" from "performances"

Line 379 - replace "has shown" with "did show"

Author Response

Reviewers suggestion: This is a very interesting paper.  Heat stress is a critical issue in terms of both production and animal welfare.  Any research that offers better management of heat stress is certainly needed.  Your pine particle work offers some valuable insight.  The Shannon index results are interesting.  Prebiotics are very popular today and this offers anther possible alternative with promising results.

Authors reply: Authors are thankful to the reviewer for providing valuable suggestions and appreciation.  The authors have addressed all the points raised by the reviewer, as shown below.

Reviewers suggestion: Line 18 - replace "has" with "have"

Authors reply: Replacement has been made as per the reviewer’s suggestion, as shown below:

Line 18: ‘Prebiotics have gained attention as potential substances for improving gut health.’

Reviewers suggestion: Line 21 - add "s" to "chicken"

Authors reply: Change has been made as per the reviewer’s suggestion, as shown below:

Line 21: ‘Heat stress (HS) has been known to have drastic effects on chickens’

Reviewers suggestion: Line 52 - delete "The" to start the sentence

Authors reply: Correction has been made as per the reviewer’s suggestion, as shown below:

Lines 52-53: ‘Broiler production contributes significantly to the livestock industry and is severely affected under high ambient temperature conditions.’

Reviewers suggestion: Line 56 - replace "encounter" with "offset"

Authors reply: Corrected as per the reviewer’s suggestion, as shown below:

Lines 55-56: ‘Various dietary supplements have been tried to minimize the drastic effects of HS.’

Reviewers suggestion: Line 57 - "the" between "keep" and "feed"

Authors reply: Correction has been made as per the reviewer’s suggestion, as shown below:

Lines 56-57: ‘However, due to the extensive competitive nature of the poultry industry, it is critical to keep feed cost at the lower end.’

Reviewers suggestion: Line 60 - replace "will" with "may"

Authors reply: Correction has been made as per the reviewer’s suggestion, as shown below:

Lines 59-60: ‘Furthermore, choosing a feed additive with heat mitigating properties may provide extra benefits.’

Reviewers suggestion: Line 100 - were these chicks males, females, or straight runs?

Authors reply: The chicks were straight run (mixed sex).  The statement mentioning ‘mixed sex’ has been included as per the reviewer’s suggestion, as shown below:

Lines 100-102: ‘A total of 260-day old straight run (mixed sex) Ross 308 broiler chicks were procured from a local hatchery (Ohsung Hatchery, Seongju, Korea) and raised in a controlled environment with continuous lighting.’

Reviewers suggestion: Line 229 - add "s" to "groups"

Authors reply: Correction has made as per the reviewer’s suggestion, as shown below:

Lines 235-236: ‘The OTUs and Chao1 were similar and were not affected among the treatment groups.’

Reviewers suggestion: Line 330 - remove "s" from "performances"

Authors reply: Correction has made as per the reviewer’s suggestion, as shown below:

Lines 343-344: ‘HS is among the environmental stressors known to have drastic effects on production performance in chickens.’

Reviewers suggestion: Line 331 - delete "the" between "on" and "production".

Line 331 - remove "s" from "performances".

Authors reply: Correction has made as per the reviewer’s suggestion, as shown below:

Lines 343-344: ‘HS is among the environmental stressors known to have drastic effects on production performance in chickens.’

Reviewers suggestion: Line 379 - replace "has shown" with "did show"

Authors reply: Correction has been made as per the reviewer’s suggestion, as shown below:

Lines 392-393: ‘However, the Shannon index did show an increasing trend in 2% HS and 0% HS treatments.’

Round 2

Reviewer 2 Report

The author has revised the questions raised, and I have no further comments.